# Understanding patient and family experiences of critical care in Bangladesh and India: What are the priority actions to promote person-centred care?

Rebecca Inglis[1]*, Meghan Leaver[1], Christopher Pell[2,3,4], Suma Ahmad[5], Shamima Akter[6], Fakrul Ibne Amir Bhuia[1], Mumnoon Ansary[7], Sidharth B. S.[5], Momtaz Begum[6], Shishir Ranjan Chakraborty[7], Hasnat Chowdhury[1], Mohammed Abdur Rahman Chowdhury[6], Putul Deb[8], Nazmin Akhter Farzana[8], Aniruddha Ghose[8], Mohammad Harun Or Roshid[8], Md. Rezaul Hoque Tipu[8], Sakib Hosain[1], Md. Mozaffer Hossain[6], Mohammad Moinul Islam[7], Bharath Kumar Tirupakuzhi Vijayaraghavan[9], Mohammad Mohsin[6], Manisha Mund[10], Shamema Nasrin[7], Ranjan Kumar Nath[8], Subhasish Nayak[5], Nibedita Pani[11], Shohel Ahmmad Sarker[7], Arjen Dondorp[1], Swagata Tripathy[5], Md. Abul Faiz[12]

1 Centre for Tropical Medicine & Global Health, Nuffield Department of Medicine, University of Oxford, Oxford, United Kingdom, 2 Amsterdam University Medical Center, Department of Global Health, University of Amsterdam, Amsterdam, the Netherlands, 3 Amsterdam Institute for Global Health and Development (AIGHD), Amsterdam, the Netherlands, 4 Amsterdam Public Health Research Institute, Amsterdam, the Netherlands, 5 Department of Anaesthesiology and Critical Care, All India Institute of Medical Sciences—Bhubaneswar, Odisha, India, 6 Department of Anaesthesia, Pain, Palliative and Intensive Care, Dhaka Medical College and Hospital, Dhaka, Bangladesh, 7 Department of Anaesthesia, Critical Care and Pain Medicine, Sylhet MAG Osmani Medical College, Sylhet, Bangladesh, 8 Department of Anaesthesiology and Intensive Care Medicine, Chittagong Medical College Hospital, Chattogram, Bangladesh, 9 Department of Critical Care Medicine, Apollo Hospitals, Chennai, India, 10 Department of Anaesthesiology and Critical Care, SCB Medical College & Hospital, Cuttack, Odisha, India, 11 Postgraduate Institute of Medical Education and Research and Capital Hospital, Bhubaneswar, Odisha, India, 12 Dev Care Foundation, Dhaka, Bangladesh

* rebecca.i@tropmedres.ac

**Data Availability Statement:** The qualitative data collected for this study contains sensitive information that could lead to the indentification of

## Abstract

Patients' experiences in the intensive care unit (ICU) can enhance or impair their subsequent recovery. Improving patient and family experiences on the ICU is an important part of providing high quality care. There is little evidence to guide how to do this in a South Asian critical care context. This study addresses this gap by exploring the experiences of critically ill patients and their families in ICUs in Bangladesh and India. We elicit suggestions for improvements from patients, families and staff and highlight examples of practices that support person-centred care. This multi-site hospital ethnography was carried out in five ICUs in government hospitals in Bangladesh and India, selected using purposive sampling. Qualitative data were collected using non-participant observation and semi-structured interviews and analysed using reflexive thematic analysis. A total of 108 interviews were conducted with patients, families, and ICU staff. Over 1000 hours of observation were carried out across the five study sites. We identified important mediators of patient and family experience that span many different aspects of care. Factors that promote person-centred care include access to ICU for families, support for family involvement in care delivery, clear

patients, staff and study sites, despite the data having been de-identified. The study team and host hospitals are also aware of the potential for reputational damage to the host institutions that could ensue. Data access requests can be made to the relevant ethics committees at the following e-mail addresses. iecsb19@gmail.com iec@aiimsbhubaneswar.edu.in info@bmrcbd.org.

**Funding:** This study was funded by the Wellcome Trust [220211], http://wellcome.org. The funder had no role in study design, data collection and analysis, decision to publish, or preparation of the manuscript.

**Competing interests:** The authors have declared that no competing interests exist.

communication with patients and families, good symptom management for patients, support for rehabilitation, and measures to address the physical, environmental and financial needs of the family. This study has generated a list of recommendations that can be used by policy makers and practitioners who wish to implement person-centred principles in the ICU.

## Introduction

The World Health Organization highlights the need to '*fundamentally recalibrate*' health services to put users at the centre of service delivery [1, 2]. The organisation's global strategy report on people-centred health services cites a range of potential benefits for patients and healthcare staff. It also recommends that strategies to promote people-centred care should be adapted to the context, whether national or regional [2]. The challenge for policymakers in many countries is that there is little or no context-specific evidence to guide the implementation of such strategies.

Person-centred care involves '*respectful communication, appropriate information sharing and shared decision-making, addressing psychological, social, spiritual and cultural needs and enhancing coordination and continuity of care*' [3]. It is particularly important for critically ill patients because adverse experiences during their admission to the intensive care unit (ICU) are known to impact their recovery [4] and that of their family [5]. Various interventions that target person-centred care have been shown to decrease ICU length of stay or improve other patient- and family-important outcomes [6]. The ICU Liberation Bundle, a raft of interventions which incorporate person-centred principles, has been shown to reduce delirium, improve functional status on discharge and improve survival [7].

The first step towards person-centred care in the ICU is to understand more about current experiences from the perspective of patients and their families. In India, studies conducted in three ICUs used a survey tool to assess family satisfaction [8–10]. However, the survey findings offer little insight into the actual experiences of families. A qualitative study in another ICU in India revealed more about the experiences of the 20 family members involved and highlighted the many stressors' families face [11]. Of note, only one of these studies was conducted in a government facility [9]; the remainder were in private ICUs. In Bangladesh, an ethnographic study of an orthopaedic ward described the challenges encountered by family members while delivering care to relatives [12] but there are no relevant studies from ICUs. Little research has focused specifically on the experiences of critically ill patients in ICUs in either country or explored the views of ICU staff on this topic. It remains unclear what factors contribute to the experiences of critically ill patients and how to support this key patient group more effectively.

In this article, we explore the experiences of ICU patients, families and staff and aim to identify person-centred practices that are feasible in publicly funded health facilities in Bangladesh and India. We highlight examples of interventions that are already used effectively in this context and draw on the insights and expertise of ICU staff. Our target audience is both clinicians and policymakers who wish to put the principles of person-centred care into practice in the complex environment of the intensive care unit.

### Research questions

We address two research questions:

1. How do patients, families and staff feel that the ICU experience could be improved for patients and their families?

2. What current practices appear to improve experiences for ICU patients and their families?

The findings described in this report are part of a larger study that focuses on the mediators of high-quality care on ICU. Other study findings will be reported elsewhere.

## Methods

### Study design

A multisite hospital ethnography was carried out in five ICUs in hospitals in Bangladesh and India. The sites were selected using purposive sampling based on their funding model (all government hospitals) and their size (all large regional units). The distance between the sites was an additional factor influencing site selection in India to enable the same research team to access both units; this was not relevant in Bangladesh as the country is smaller. Government hospitals were chosen to allow a specific focus on public facilities, with the goal of producing study findings that are relevant to guide government policy. Large regional ICUs with bed numbers of 12 or above were chosen to make comparisons between the sites more meaningful.

All five ICUs treat a mix of medical and surgical adult patients. They all follow a predominantly closed ICU model, meaning the intensive care team was primarily responsible for patient care and decisions about who to admit. At the request of the centres involved, no further information regarding the sites will be provided to protect confidentiality. This was requested as a condition of participation, given the sensitive nature of the data collected and the perceived potential for reputational damage to the host institutions that could ensue.

### Data collection

Data were collected through observation and semi-structured interviews between 01/02/22 and 31/12/22. The data collection team comprised eight researchers: four clinician researchers (authors MARC, NAF, MA and SB, all doctors) and four social scientists (authors SH, FIAB, HC and SN). At each study site, one clinician researcher and one social scientist worked together to collect the data. Author RI provided in-person and remote supervision during data collection and collected additional observational data from all five sites. RI is an ICU doctor with experience of ethnographic research methods. Data collection was overseen by senior clinicians MAF and ST as the national study leads for Bangladesh and India respectively, while authors CP and ML provided expertise remotely.

The target number of interviews and the approximate duration of observations in each ICU was determined prior to the start of data collection using the concept of 'information power' [13]. These estimates were revised during data collection based on regular appraisals of the adequacy of the data collected to address the research questions [14]. Data adequacy was assessed during weekly team discussions. The final number of interviews conducted and the duration of observations in each site is given in Table 1.

**Observations.** The observations focused on the activities of staff in the ICU, including direct patient care activities, and interactions with patients' families. The observations were unstructured.

The degree of participation during observation varied between researchers. One clinician researcher continued to work clinically while conducting observations. Two other clinician researchers had previously worked clinically in the ICU where they were collecting data, but not at the time of the study. The remaining clinician researcher and all the social scientists adopted a non-participant role during observation. Only one clinician researcher had no

**Table 1. Details of data collection at each site.**

| Site | 1 | 2 | 3 | 4 | 5 | Total |
|---|---|---|---|---|---|---|
| Data collection timeline | February 2022 –June 2022 | March 2022 –June 2022 | May 2022 –December 2022 | May 2022 –December 2022 | September 2022 –December 2022 | |
| **Observations** | | | | | | |
| | 347 hours | 160 hours | 176 hours | 75 hours | 335 hours | **1093** |
| **Interviews** | | | | | | |
| **Junior nurses:** | 4 | 4 | 5 | 4 | 4 | |
| **Senior nurses:** | 4 | 4 | 4 | 5 | 4 | |
| **Junior doctors:** | 4 | 4 | 4 | 4 | 3 | |
| **Senior doctors:** | 4 | 4 | 4 | 4 | 4 | |
| **Patients:** | 3 | 3 | 3 | 0 | 3 | |
| **Caregivers:** | 4 | 4 | 3 | 0 | 4 | |
| | 23 | 23 | 23 | 17 | 22 | 108 |

'Junior nurse' denotes a nurse with less than 3 years' experience on ICU. 'Junior doctor' denotes a doctor who has not completed their ICU training.

previous ICU experience, so they were supported by senior intensivists SA and MM who provided technical support where required.

The researchers' differing roles, degree of participation during observation, and positionality all influenced the data that was collected. Data collection was organised in such a way that observational data was often collected from the same event in tandem by two researchers with different professional backgrounds (one physician and one social scientist) so these influences could be interrogated and were discussed frequently. The resulting insights added depth to the data.

Observation data were captured in the form of contemporaneous hand-written field-notes that were subsequently expanded into typed notes by individual researchers. Data were collected at different times of day, including night shifts. Between 75 and 347 hours of observation were carried out on each ICU.

**Interviews.** Interviews were conducted with patients and patients' families in four out of five of the ICUs. Patients were invited for an interview if they met the following criteria: a) ready for discharge from ICU, b) deemed able to give informed consent by the treating team, and c) able to communicate verbally. A member of the patient's family was invited for an interview if their relative was clinically stable and ready for discharge from ICU. Patients and family members who met these criteria were sampled consecutively. Nurses and doctors were offered an interview using a maximum variability strategy based on role and seniority.

The interviews were semi-structured and focused on the interviewee's experience of receiving or giving care on the ICU (see 'S1 Text'). Interviewees were asked to reflect on ways in which the experiences of patients and their families could be improved. The interviews were conducted in the preferred language of the interviewee and were audio recorded. In Bangladesh, all interviews were transcribed in Bengali and then translated into English. One researcher reviewed all the transcripts and translations for quality control. In India, interviews conducted in Hindi or the local language were translated directly into English by the two trilingual interviewers, crosschecking with each other when there were Hindi translation queries. A third person was consulted for queries in the local language. It was possible to resolve all translation queries in this manner. The interviews conducted in English were transcribed only.

**Consent.** Prior to starting data collection in each site, the research team held a series of meetings with the ICU staff to explain the purpose of the study. The meetings were staggered

to allow staff working different shifts to attend. The content of the meeting was structured around the information on the participant information leaflet and attendees were given a copy of the participant information leaflet in a language of their choice. Staff were invited to ask questions during the meetings or to speak to a member of the research team privately afterwards; for several days after the meeting the researchers made themselves available in a private area of the ICU for this purpose. It was emphasised that involvement in the study was entirely optional. Two members of staff across the five sites initially declined to take part; both changed their mind and asked to get involved part way through the study.

After completing the preliminary steps of the consent process outlined above, verbal consent was obtained from staff at the start of each period of observation. Verbal consent was also obtained from family members when they were present during the observation. Although not the direct focus of the observations, patients were made aware of the presence and purpose of the research team whenever their condition allowed. This was not possible in many cases as they were unconscious or sedated.

Written consent was obtained prior to all interviews. Adapted versions of the participant information leaflet were used for patients and family members. During the consent process it was emphasized that the decision to participate or not would not influence the medical care they received.

**Ethics statement.**   The study received ethical approval from the Bangladesh Medical Research Council, the appropriate institutional ethics boards in India, and the University of Oxford. A study protocol amendment to include interviews with patients and family members also received ethical approval.

## Data analysis

A reflexive thematic analysis approach was used to address the research questions [15]. Our team adopted a constructionist epistemology with an experiential orientation to data interpretation, underpinned by a realist ontology [16]. The analysis was informed by concepts from the person-centred care literature, including the Picker Principles of Person-Centred Care [17] and the global practice-based framework of person-centred care by Giusti *et al*. [18]. NVIVO 1.7 software (QSR International, Australia) was used to manage the data.

Weekly meetings were held online to discuss the data as it was collected. The India and Bangladesh teams met separately but authors RI and CP attended both meetings to enable data sharing and comparison between all sites. Preliminary findings requiring clarification or further exploration were identified and used to inform subsequent data collection.

Once data collection was complete, RI examined the full dataset and highlighted data that broadly pertained to patient or family experience. These data were then open-coded based on semantic meaning. Preliminary themes were created from the coded data in a recursive manner, going back to refine the coding where needed. As the analysis progressed, a more deductive approach was used to ensure that the themes and subthemes being generated were of relevance to the research questions. During the later stages of analysis, research team members who were not directly involved in data collection reviewed the preliminary themes to sense-check and probe the theme descriptions, to enhance reflexivity and analytical depth.

The recommendations to promote person-centred care in Table 2 were derived from the main mediators of patient and family experience identified during analysis. The recommendations either describe practices that were seen to be successful in one or more study site (e.g., supporting nurses to update families) or directly address a common and important problem identified (e.g., no safe spaces for women). Suggestions from patients, families and staff were taken into account when compiling the list. All authors reviewed the feasibility and

**Table 2. Recommendations for promoting person-centred care on ICU.**

| Study finding | Suggestions for implementation |
|---|---|
| **1. Supporting family involvement on ICU can promote person-centred care** | ⇒ Consider less restrictive ICU visiting for selected family members so they can spend more time at the bedside and play a greater role in patient care.<br>⇒ Provide all families with a clear induction to ICU, with a member of staff explaining how they can support their relative while on ICU and what rules they need to follow.<br>⇒ Develop a scope of practice to describe what tasks family members are allowed to perform. Train them how to perform these tasks and provide close supervision where needed.<br>⇒ Ensure families are taught how to follow infection prevention and control measures and are provided with the necessary infrastructure (e.g., sinks) and supplies (e.g., soap, alcohol hand rub, paper towels) to comply. |
| **2. Communication is a key mediator of patient and family experience** | ⇒ Support nurses to communicate routine updates to patients and patients' families as deemed clinically appropriate. Consider discussing what information the nurse can relay to the family each day on the ward round.<br>⇒ Where possible, allocate the same nurse to a patient for consecutive days to allow them to build rapport with the patient and family.<br>⇒ Ensure that senior doctors are routinely involved in complex discussions and that junior staff are well supported when communicating with families.<br>⇒ Consider establishing a regular, dedicated timeslot for senior staff to communicate with each family, or a system where families can sign up for a communication slot with senior staff.<br>⇒ Implement a communication proforma, where important conversations with the patient or family are documented. File it in the patient record, to improve continuity of communication.<br>⇒ Establish a dedicated location for communication with families, ideally close enough to the ICU that the patient could join in the conversation when possible.<br>⇒ Discuss key management decisions (e.g., initiation of mechanical ventilation) and likely eventualities (e.g., end-of-life planning for a patient unlikely to survive) pre-emptively to avoid the challenge of contacting family members at short notice.<br>⇒ Develop a robust system to collect all family contact details for when there is a need to contact families at short notice. |
| **3. Symptom management, rehabilitation, and reducing environmental stressors should supplement curative care on ICU** | ⇒ Encourage proactive pain management with regular pain scoring, protocolised pain management and reliable availability of morphine and fentanyl (or equivalent drugs).<br>⇒ Consider making physiotherapy and mobilisation nurse-led and part of the care delivered to every patient on ICU. Explore how to involve the family in delivery. A sturdy bedside chair where the patient can sit once able is a key item of equipment.<br>⇒ Decrease extraneous noise by regularly servicing and repairing ICU equipment to avoid constant alarming. Encourage families to bring in headphones and music or radio for patients where appropriate. |

*(Continued)*

**Table 2.** (Continued)

| Study finding | Suggestions for implementation |
|---|---|
| **4. Addressing the needs of families can improve patient care** | ⇒ Establish a waiting area with facilities for families, including a dedicated space for women. Prioritise handwashing facilities, washrooms and access to clean water for drinking. An electrical socket with a multiplug adaptor to charge mobile phones is helpful. <br> ⇒ Highlight potential sources of financial support to allcomers on the ICU and support people to access them. <br> ⇒ Decrease hassle for families when procuring drugs and other supplies by minimising the number of requests, giving them precise instructions where to go and maximising the use of government supplies. <br> ⇒ Find workarounds to streamline or eliminate the role of the family in procuring investigations. <br> ⇒ Minimise avoidable ICU expenses by ensuring a steady supply of government-funded provisions, prescribe using generic drug names where appropriate, and avoid non-evidence-based treatments. <br> ⇒ Explore local solutions to minimise stress and aggravation for patients and families caused by hospital support staff and the payments they request for their services. |

acceptability of each suggestion to generate the final list. This report adheres to the Standards for Reporting Qualitative Research (SRQR) guideline [19].

## Inclusivity in global research

Additional information regarding the ethical, cultural, and scientific considerations specific to inclusivity in global research is included in the S1 Checklist.

## Results

The themes and subthemes generated during analysis are summarised in Fig 1.

## 1. Supporting family involvement on ICU can promote person-centred care

**1a) Less restrictive access to ICU can improve patient and family experience.** Across the five ICUs, the amount of time that family members were allowed to spend at the bedside ranged from restrictive, with admission to ICU only permitted when someone was needed to carry out a task, to liberal, with family permitted to enter at almost any time of day. Restricting families from visiting their relative on ICU was a frequent cause of distress and was highlighted by many patients and family members.

*The most difficult thing about being in the ICU is that we have to be separated from the patient. [. . .] I was upset about that, not being able to be beside my patient's side in her bad condition, I had to stay out. (Interview with a family member, F1)*

*Being alone is the worst. It would be better if we could be together. When I'm sick, I don't like being alone. (Interview with a patient, P1)*

In practice, it was noted that families were often permitted to access ICU outside of the official visiting hours, depending on which members of staff were on duty. Most patients and families–and even some members of staff–were unsure exactly what the rules around visiting

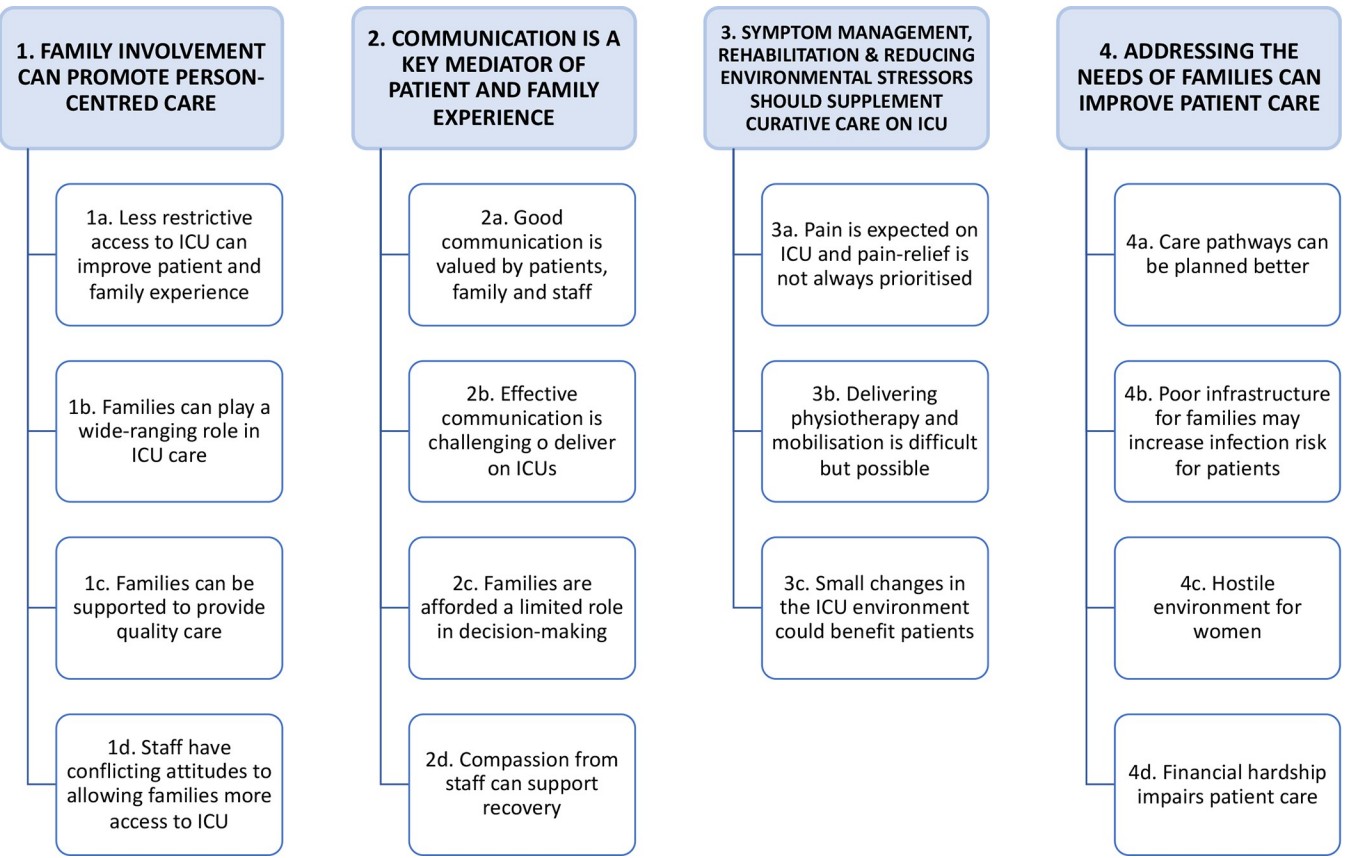

**Fig 1. Themes and subthemes generated during data analysis.** Themes are shown in blue boxes and subthemes are shown in white boxes.

hours were. The senior doctors' morning ward round was the time of day when family members were most strictly excluded, to the frustration of some families:

*Family member: Whenever doctors come for rounds, they don't call us. We may have a lot of questions to ask. But we never got a chance [. . .].*

*Interviewer: You think relatives should be allowed to remain at the bedside during rounds?*

*Family member: Yes. (Interview with a family member, F2)*

**1b) Families can play a wide-ranging role in ICU care.** In the ICUs with flexible visiting policies, families were more able to contribute to patient care in ways that went beyond fetching supplies and delivering food. Family members were observed to perform a wide range of additional roles, as summarised in Fig 2. As well as direct patient care and logistical support, family members were found to contribute to patient safety and rehabilitation. By contrast, in ICUs with more restrictive visiting policies, families were mainly expected to wait outside the entrance to ICU, from where they could be summoned when needed.

Families reported the psychological benefits of being able to provide care for their critically ill relative:

*I was able to help serve my sister. . . the experience of serving her was good. (Interview with a family member, F3)*

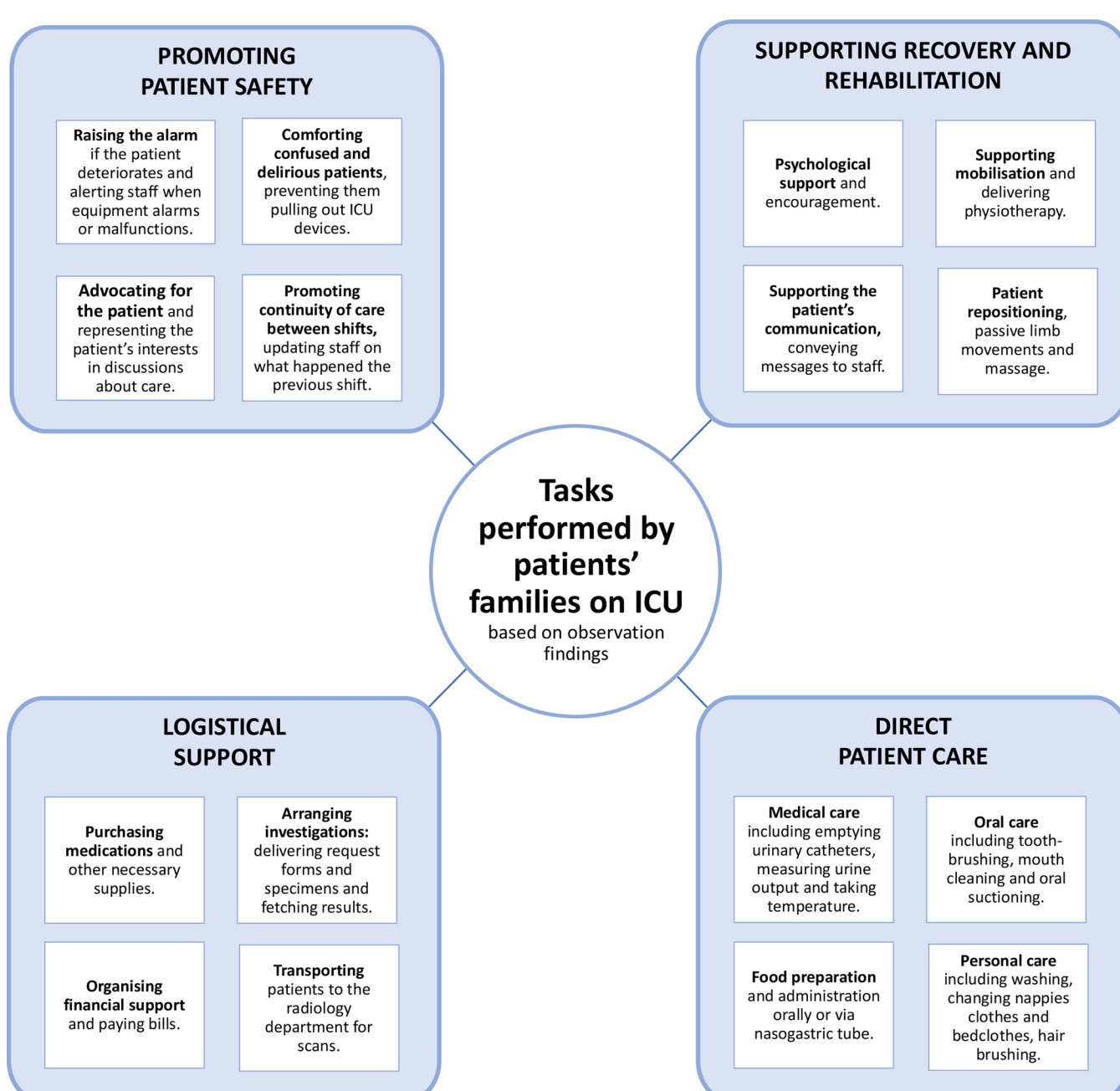

**Fig 2. The tasks that ICU patients' families were observed to perform.**

Several family members expressed a desire do to more direct patient care, such as this mother whose son was admitted to an ICU with restrictive visiting.

*Interviewer: Have you been able to help provide care for your relative?*

*Family member: Yes, only during feeding and brushing teeth. The nurses didn't allow me [at first], but I requested to do so. (Interview with a family member, F4)*

Her son became confused and agitated while receiving mechanical ventilation, so the staff tied straps from his wrists to the bedframe to stop him pulling out his endotracheal tube. The interviewee described how distressing she found this:

*My son's hands were tied when he was admitted to ICU. As a mother, I didn't tolerate this. But later I understood why it was essential for safety and care. (Interview with a family member, F4)*

By contrast, when relatives were allowed freer access to ICU, they could sometimes help to settle agitated patients, so staff did not always have to resort to physical restraint. One respondent described how she was able to support her father when he became delirious:

*My father become too excited. It became difficult to manage when he threw his hands and feet around. He wants to go home, doesn't want to stay here. Then he calms down when he sees us. Then I massage his head a little, massage his feet a little. . ." (Interview with a family member, F5)*

As well as decreasing the use of physical restraint, when family members were present at the bedside, staff were sometimes able to reduce the amount of sedation that patients were receiving, potentially helping to get patients off the ventilator sooner.

*Pt [patient] was in ventilator, after meeting with her family member she became quiet and gripped her husband tightly, that she is alive, her husband also cried so much. [. . .] After meeting with her family member, we reduced the dose of sedation, and ultimately early extubation may be done. (Study team observation)*

When they were permitted to spend longer periods at the bedside, families were also observed to play another important role: they would raise the alarm when their relative deteriorated.

*His face was suddenly dry and different. When I gave him water, he could not swallow the water. He started having trouble breathing, he gestured to me and said he felt bad. Then I hurriedly called the nurses. The nurses came running to my son. Came and measured the [blood] pressure. The nurses called the doctor. The doctors came quickly and gave my son a mask and oxygen. (Interview with a family member, F6)*

Family members were also observed alerting staff when the monitor or the ventilator alarmed. Some families who had been in ICU long enough to understand normal and abnormal vital signs would even alert staff when they saw a deterioration.

*In the case of my sister, I saw that the electrical supply of the machine was disconnected. Then the authorities took action. The machine was running, but the battery backup was lost, it was like that for 5–10 minutes, which is harmful to a critical patient. (Interview with a family member, F3)*

*He had a pipe in his throat. He used to only speak in gestures I didn't understand. I would just ask, should I turn off the fan, he would say 'yes', meaning nod his head, I would just understand. I would say, do you have a stomach ache? If he nodded, I would tell the doctor. (Interview with a family member, F7)*

As well as alerting staff to actual emergencies, there were also instances when family members were observed alerting staff to less urgent matters, such as a disconnected oxygen saturation probe or an infusion that was complete. Without being told which changes were most concerning, it was hard for them to distinguish what was most urgent.

**1c) Families can be supported to provide quality care.**   Common tasks that family members were asked to perform included repositioning, feeding or mouth cleaning. These were made challenging by the unfamiliar ICU paraphernalia so close supervision was required. Families also needed to be supported to follow infection prevention and control measures.

*They said, look, everyone is busy with work, if you position your sister on a tilt and move her every 3 hours, she won't get bed sores. Clean her properly twice a day. I said okay. But it was difficult for both of us to do the work because there is the CV line, the oxygen tube in her mouth, the oximeter on her hand and the BP cuff [. . .] but now I have learned how to move her. (Interview with a family member, F3)*

The time individual members of staff spent teaching and overseeing the care delivered by families varied greatly. As a result, so did the quality of the care they were able to provide. On occasion, family members were observed to deliver food too fast via the patient's feeding tube or to feed the patient while they were lying flat rather than propped up. When relatives were expected to wear single-use hospital gowns to enter the ICU, these were often reused repeatedly rather than disposed of as intended. Nonetheless, there were many more occasions where families were observed to carry out tasks correctly with care and attention.

**1d) Staff have conflicting attitudes to allowing families more access to ICU.**   In contrast to the importance placed on ICU access by patients and families, many ICU doctors felt that families should have less access to ICU. 'Visitor restriction' was frequently suggested as a way to improve the quality of care on ICU.

*And if the security [. . .] was increased to 24 hours, it would be better and the patient attendants would not be constantly coming and going in the ICU. Our service quality would improve, cross-infections would get much fewer, and we could keep the ICU clean. (Interview with a senior doctor, SD1)*

The conviction that families were a source of infection on ICU was widespread across all five ICUs:

*There should be some restrictions for patients' relatives to our ICU [. . .] We are getting lots of sepsis here, as too many people are coming inside and touching. (Interview with a senior doctor, SD2)*

Although many nurses expressed similar concerns about patients' families being 'unhygienic', they conceded that families played an indispensable role in the provision of patient care.

*We say it ourselves: in fact, ICU patients cannot be cared for alone. Help is needed here. [. . .] If [the family] help, the patient improves. (Interview with a senior nurse, SN1)*

*Without them, delivering care would have been challenging. (Interview with a senior nurse, SN2)*

*We do not have a sufficient number of staff to run the ICU perfectly. There are not enough attendants to assist us, so we advise patient families to come and help us with positioning,*

*changing diapers, brushing teeth, feeding, and washing the face of their patient. (Interview with a junior nurse, JN1)*

Many nurses said that families helping with patient care made their job easier by decreasing their workload. This view contrasted with reports from other nurses and doctors, who felt that family members being allowed on ICU created more work for them because the families wanted to ask questions or disrupted their workflow in other ways.

*When a patient comes to the ICU, their family is actually the biggest source of tension [. . .] In the ICU there are actually so many patients, so much stress, that we cannot always talk with the patient's party. (Interview with a junior doctor, JD1)*

*The attendant of bed 2 asked the doctor about his patient. He also brought the patient's file. The doctor said, I don't know anything about your patient. You will know your patient update every morning. I have neither seen nor treated your patient. (Study team observation)*

One clinician researcher in our team reflected on how her attitude to patient access on ICU changed during the course of the study:

*Before working in this study, like others, I also thought that restricting visitors in ICU will be most beneficial for patient outcome. But, after working with you and taking interview from patient and patient's caregiver, I realized that if the patient's caregiver always accompanies the patient and takes part in the patient's care it makes the recovery smoother. It gives great psychological support to patient. Also, the caregiver can see the patient's progress day by day, she/he can get the answer to their queries easily due to easy access to the doctors and nurse. (Study team reflection)*

## 2. Communication is a key mediator of patient and family experience

**2a) Good communication is valued by patients, family and staff.**   Communication was repeatedly identified by staff, patients and family members as a key mediator of satisfaction with ICU care. However, there was also widespread acknowledgement from staff that current communication practices need to be improved.

*Personally, I feel like the communication between the doctors or the nurses and the attendants should be good, [but] we sometimes are, like, so overworked, so overstressed, we do not communicate in a way that we should communicate with them, we do not pass the information that they should know. It's a really stressful atmosphere and that makes the communication. . .[pause]. . . it should be better. (Interview with a junior doctor, JD2)*

*'If we could talk to caregivers, listen to their experiences, spend a little more time talking about patient improvement, their experience would be better. We basically don't have time to talk. I behave rudely, it happens because of pressure or because of their misunderstanding.' (Interview with a junior doctor, JD3)*

Staff often viewed communication as something that was in their power to improve, unlike other mediators of patient and family experience such as infrastructure.

*Interviewer: How can we improve the experience of families whose loved ones are in the ICU? Describe what you can think of.*

*Doctor: Responsibilities and duties fall within us to satisfy loved ones. We can do that through counselling. (Interview with a senior doctor, SD1)*

Some members of staff appeared to regard communication as an interruption to the work that they felt they should be doing.

*Doctors are always busy. They cannot talk unless it is necessary. [Patients' families] should understand that doctors are very busy, sisters [nurses] are busy. They cannot be disturbed. (Interview with a junior doctor, JD1)*

Nevertheless, most staff interviewed placed a high value on communicating with the patient and their family.

**2b) Effective communication is challenging to deliver on ICUs.** Across all five ICUs, very few of the patients or family members interviewed were able to demonstrate an understanding of their diagnosis, their treatment, or how their condition was likely to progress. While this might reflect deficiencies in communication practices, many patients and families also said that they felt satisfied with the communication they received from staff.

*What I needed, what I needed to say, what I felt that it is important to say, I asked. (Interview with a family member, F8)*

Specific challenges that the study team identified in the communication process are listed in Fig 3.

Several practices were observed that appeared to promote better communication with patients and families. For example, in one study site they had established a regular timeslot three times a week when a senior clinician would update the families of every patient on ICU:

*The nurses prepared a list of all the patients which summarised their diagnosis and treatment plan. One representative of each family was summoned and asked to line up outside the office.*

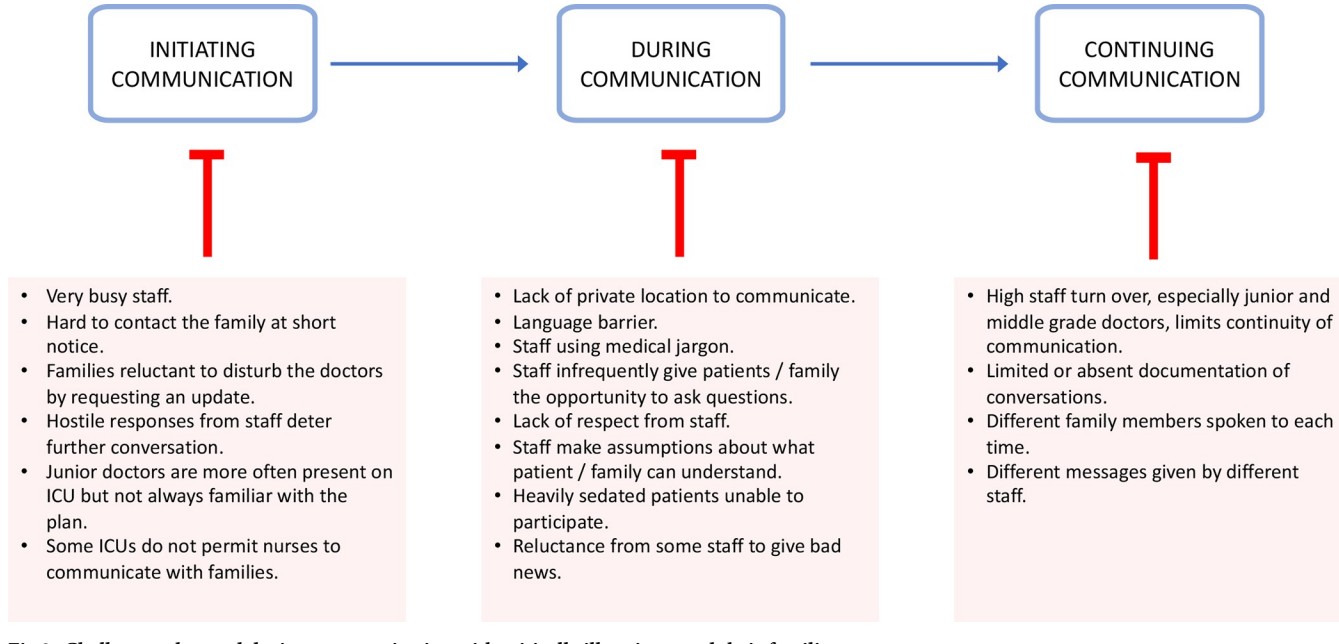

**Fig 3. Challenges observed during communication with critically ill patients and their families.**

*The head of ICU sat in the office and updated each person, one after the other. Each interaction lasted about five minutes and the communication was predominantly unidirectional, with the doctor doing almost all of the talking. The family member seemed to have the chance to ask one question and no more before the next person was beckoned in. (Study team observation)*

Based on our observations, this scheduled update meeting did not allow for detailed communication or give an opportunity to address family questions or concerns in full, but it did ensure that all families received an authoritative update on a regular basis.

Another seemingly successful practice in two other ICUs was the involvement of nurses in communicating with patients. Nurses on these units were permitted to communicate routine status updates to the families, freeing up the doctors to prioritise communication of major changes only.

*Our senior sirs have taught us that our senior nurses–due to their long practical experience– call and communicate with the patient's family. They also do counselling about the patient. (Interview with a senior doctor, SD1)*

This approach capitalised on the fact that patients and family members often described nursing staff as more approachable and easier to understand than the doctors. Each nurse also cares for a smaller number of patients than the doctors so they often know them in more detail. Where nurses were allocated to the same patient on successive days, they were also able to build up rapport with the family, improving the continuity of communication.

Nurses in other ICUs where nurses were not permitted to update patient families expressed a desire to move towards a similar system:

*I personally love counselling patient parties [. . .] In our team, the doctor is supposed to do the counselling. We are told we cannot do counselling. [However] families are more satisfied when they are kept up to date with patient information. (Interview with a senior nurse, SN3)*

Other practices that appeared to aid communication included: a communication form to add to the patient notes to document important conversations with the family; a dedicated room to communicate with privacy; and ensuring senior staff were involved in communicating with families. Higher staffing levels also appeared to have a positive influence on communication.

**2c) Families are afforded a limited role in decision-making.**   Patients' families were rarely seen to be included in decisions that involved weighing up the relative merits of different management strategies. Much more commonly, they were asked to provide consent for an invasive procedure or for an expensive treatment that was not covered by government funding.

In one interview, a senior nurse described the process of obtaining a family's permission after the doctors decided that a patient should be put on a ventilator.

*[The family] are told that the patient may be in danger, and we want them to get well. But, God forbid, if they get worse, then there is nothing we can do. [. . .] The patient party wants the patient to survive. The patient may get better or worse. Consent is taken so that we will not be responsible if they get worse. (Interview with a senior nurse, SN4)*

As the nurse clearly articulates, the reason for taking consent was to absolve the hospital of responsibility in the event of the patient dying rather than to have their involvement in

decision making. Indeed, when families wanted to ask more questions about the treatment plan, several described how staff were reluctant to engage.

*They become bored if I ask more than one question. (Interview with a family member, F9)*

Some time-critical communication was limited by the challenges of getting hold of family members at short notice. Families were usually contacted via a public address system, by calling their mobile telephone or by sending someone to find them outside the front of ICU.

*We don't have any public address system inside ICU to call patients' relatives. We have to come to the doorstep and inform the security guard to pass the information to the families. This process usually takes 3 to 5 minutes which I never liked. The introduction of an public address system will definitely help us. We simply make an announcement, and the process will be seamless. There is no proper waiting area for patient families. They spend their days sitting on the staircase outside ICU. [. . .] We sometimes fail to find patients' families outside the ICU when we have to pass on some information, or we have to take consent for some procedures like central line, intubation, etc. Sometimes their mobile phones are unreachable. If there is a public address system and a waiting lobby, information or a message will be delivered smoothly. (Interview with a senior nurse, SN2)*

**2d) Compassion from staff can support recovery.** Many patients and families spoke very warmly about the impact that kindness from the staff had on their experience of ICU.

*The nurses were well behaved, spoke encouraging words, and so were the doctors. They used to take extra care. It felt so good. Patients get better with encouraging words. (Interview with a patient, P2)*

*The doctors and nurses are so good that their words would cure most of your pains and sufferings. The way they take care, I have never seen in any hospital. All credit goes to them. (Interview with a family member, F10)*

Others spoke with equal emotion about the negative impact of unkind words. For example, one young man, who was caring for his cousin on an ICU, dwelt on an exchange with a doctor from a few weeks earlier.

*Interviewer: After arriving [on ICU], what was hardest?*

*Family member: A doctor said that there is discipline here. When I saw that my patient's urine bag was full, I told him so. Then the doctor said "there is a system, there are rules to follow in the ICU. Don't try to teach us." That's it, I'm new, I don't understand these things, I'm also new to the hospital.*

*Interviewer: Did he speak with a little anger?*

*Family member: Yes.*

*Interviewer: Do you feel hurt?*

*Family member: [Laughs] Obviously.*

*Interviewer: Do you feel bad for him talking angrily?*

*Family member: Yes.*

*Interviewer: Would it have been better to say it in a softer tone?*

*Family member: Yes. Because I am new. (Interview with a family member, F8)*

During observations, staff were frequently seen scolding patients or their families. Nevertheless, during interviews, staff also seemed very aware of how they could improve their interaction with patients if they had the time and the motivation. A senior nurse was asked how to improve the experience for patients on ICU:

*If we go and talk to him, when giving chest physiotherapy, we pat his forehead and say "how is uncle? We will now give you chest physiotherapy. You will feel pain, then you will cough, we will suction it, along with urine". If I talk to them in this way, then the frustration they feel will not be there anymore. It ought to be done as though he is my family member. (Interview with a senior nurse, SN5)*

## 3. Symptom management, rehabilitation, and reducing environmental stressors should supplement curative care on ICU

**3a) Pain is expected on ICU and pain-relief is not always prioritised.**   Staff, patients and families viewed the ICU as '*a place of suffering*' (interview with a patient, P3), where physical discomfort was to be expected. Patients mentioned pain when describing their experience of ICU, yet, when asked how their experiences could be improved, few expected or requested improved pain management. One relative recalled alerting a doctor that her sister was suffering discomfort from her endotracheal tube, only to be told that discomfort was to be expected and no action was taken:

*Then the doctor said, "the tube is newly inserted, so she will feel uncomfortable if she moves". Then I understood. I do not say anything unless necessary. (Interview with a family member, F3)*

With regard to the systems and processes for managing pain, only one ICU had routine pain assessments for patients, where the nurses were expected to carry out regular pain assessments for all patients. In the other ICUs, pain management was carried out in an *ad hoc* fashion and most often focused on those patients who were able to vocalise their pain. In one site, pain management was severely hindered by the lack of intravenous morphine or fentanyl for pain relief. Another site had an inconsistent supply of these drugs. The problem was variously attributed to concerns regarding the potential for opiate/opioid abuse, supply chain problems, and regulations governing its use.

In some ICUs, patients' pain was observed to be frequently managed with sedative medication in the first instance rather than the recommended strategy of treating pain first then adding sedation if required.

*The patient is delirious due to pain, throwing their arms and legs and then we tied the arms and legs, gave sedation. (Interview with a senior nurse, SN6)*

Sedation was also described as being given with the intention to improve patient's experience of ICU.

*Interviewer: What things do you think could improve . . .[patients'] experience?*

*Doctor: There is no other way but to sedate. (Interview with a junior doctor, JD1)*

When staff were asked for their reflections on pain management in their ICU, it became clear that treatments to cure the patient were explicitly prioritised over symptomatic management by some:

*Our priority is for the patient's life but not for their comfort. As we develop, we have to start thinking of the patients comfort also. [. . .] First thing first, I think we should prioritize the patients' safety first and only then comfort. (Interview with a senior doctor, SD3)*

**3b) Delivering physiotherapy and mobilisation is difficult but possible.** Supporting a critically ill patient to regain strength with physiotherapy and helping them to get out of bed when possible is important for their recovery [20]. However, there was relatively little physiotherapy or mobilisation observed in many of the ICUs in the study. When asked about this at interview, there was a common perception that patients were too sick to mobilise:

*Mobilisation doesn't work like that here. I have seen that the existing protocols for thrombosis prevention do not work well [. . .] But it can be seen here, the condition of the patients is so bad, no one cares about picking them up or sitting them down. (Interview with a junior doctor, JD4)*

Furthermore, mobilisation was seen as a short-term risk rather than an intervention with long term benefit to the patient.

*I was told: "don't move, you are an ICU patient. If you fall, it will be a problem. Hands, feet and head; if you break them, then there will be another problem. Tell us if you need anything, if we can't, tell your attendants. They can, but don't you try to move." . . . [It is a problem] lying down all the time–it is not possible for a human being. It is possible for people who are unconscious, they are in a lying position–not in a sitting position and they do not even know it. But I am conscious, so I know, it is not possible to lie down all the time. (Interview with a patient, P4)*

Nevertheless, one ICU was a clear positive outlier with regard to mobilising patients and had successfully established a culture that valued rehabilitation. Throughout the observation period, patients would consistently be seen sitting out of their beds or mobilising with the support of staff. Their success appeared to be multifactorial. First, it was the only site where physiotherapy was predominantly nurse-led and where nurses consistently mentioned mobilising patients as part of their role. The nurses took responsibility to proactively identify patients who were ready and would ask the doctors or the nurse in charge for permission to mobilise them. Second, despite no formal training in physiotherapy, they copied what they had seen a professional physiotherapist do and aimed to get patients moving at least twice a day. Third, they also enlisted the support of patients' families, teaching them what exercises to do.

Fourth, the nurses interviewed in this ICU all saw a clear reason for mobilising the patients–they mentioned improving patient wellbeing, returning them to 'normal life', and preventing delirium–which motivated them to ensure it was done. One junior nurse described mobilisation and physiotherapy as '*basic and essential*' for patients, although acknowledged it was done better on days when they were better staffed.

A senior nurse said:

*I think [mobilisation] is better than a medicine on ICU, [it] helps the patient to come out from the depression and all.*

Fifth, the senior doctors also recognised the value of mobilisation and would regularly ask about it in on ward rounds. They later introduced a ward round checklist that included a box to tick to check that mobilisation had been done. Finally, with regard to infrastructure, this was the only ICU in the study with chairs by the bed that were suitable for ICU patients to sit on.

**3c) Small changes in the ICU environment could benefit patients.**   In the all the ICUs, there was a cacophony of alarming machines, bright lights and noisy conversation. This environment was often disturbing for patients and families alike.

*The sounds were sometimes intolerable. (Interview with a patient, P2)*

*The noises were making my heart beat faster. These noises are really scary. (Interview with a family member, F10)*

*There was a feeling of a countdown. I used to run outside the ICU and start crying. (Interview with a family member, F4)*

The patients spoke of headaches and sleep disrupted by shouting and other noises while families found it stressful not knowing why the machines were alarming. As with symptom management and mobilisation, many members of staff felt that this was an inevitable part of being on ICU:

*Interviewer: What do you think is experience is like for people who are admitted to the ICU as patients?*

*Junior doctor: As patients? Personally, if I would have been in that bed, the experience wouldn't have been good, the constant beeping of the... you know, that's really... many patients do get psychosis here, bed number four, she has ICU psychosis, another patient he had ICU psychosis because of the constant beeping and constant alarms, it's really challenging for the patients as well.*

*Interviewer: What can be done to make their experience better?*

*Junior doctor: Better? I don't think anything, ICU is a place where you cannot expect things to be better. There will be beeping of the monitors, there will be alarms, there will be sick patients by your side, you cannot help with that. I suppose their family can comfort them–and the doctors as well–but with the surroundings nothing much can be changed (Interview with a junior doctor, JD2)*

Despite the fatalism from some respondents, we observed two different strategies to address the constant noise. The first, usually initiated by the patient's families, involved bringing in headphones and a device to play music or radio to the patient. The second strategy was to repair or replace the broken machines that would alarm continuously, leading to a small improvement in unnecessary noise.

Another source of distress for patients and their families was the lack of privacy, particularly when they witnessed patients around them dying at close hand.

*When I recovered, I saw a dead body, seeing that I got scared. (Interview with a patient, P5)*

All five ICUs attempted to address this problem with screens or curtains to pull around the bed when a patient was undergoing a resuscitation attempt, during a procedure or when the patient was undressed, but these were used inconsistently in some sites. One measure that

appeared to promote more consistent use of screens or curtains during resuscitation was allocating the responsibility of ensuring privacy to the housekeeping staff. When this was left solely to the nurses and doctors, they were sometimes so focused on the clinical management that they forgot.

## 4. Addressing the needs of families can improve patient care

**4a) Care pathways can be planned better.**   The delivery of care in the study ICUs was observed to be very dependent on logistical support from patients' families. They are required to ferry investigation requests, patient samples, receipts, and test results back and forth to various laboratories, scour local pharmacies to source long lists of drugs and equipment, and fulfil the administrative requirements of the hospital. Many of these tasks involve lengthy queues and navigation around sprawling and poorly signposted hospitals and their unfamiliar environs. Families were often given requests one-by-one rather than all at once, meaning multiple trips. All the while, family members were also required to be on hand in case they were needed on ICU.

*From here they used to give many tests, to be taken to different places I did not understand. Where it should take only 5 minutes, it took me 1 hour or half an hour. (Interview with a family member, F11)*

Many nurses and doctors emphathised with the '*hassle*' that families had to endure and highlighted the clinical consequences of the delays to patients' care that resulted. These included delays in administering critical medications, families being unable to track down the correct medication, and test results being delayed or even lost.

An initiative from a senior doctor in one of the ICUs provided a successful solution. She described how many of the microbiology test results were delayed or went missing and this was compromising clinical decision-making. In response, she allocated a member of staff to take all the specimens to the microbiology lab directly rather than handing them to the families. The same person fetched any completed test results at the same time. The doctor also established a direct line of contact with the microbiology department: photographing the list of culture specimens collected that day and sending it to the microbiology resident via WhatsApp. That way, whenever one of the cultures flagged positive, the resident could contact her immediately to let her know. This approach relieved the families of one of their many tasks and resulted in a much more efficient transfer of time-critical clinical information, benefitting patients.

There were some other examples of good practices that we observed or were told about, yet many of these relied on individual- rather than systems-level solutions. These included nurses who would clearly explain to families which pharmacies stocked which items and support staff who showed families exactly where to go within the hospital. One hospital pharmacy stocked much of what was needed on ICU. Finally, certain prescribing practices could help families, for example cohorting requests, minimizing requests and generic prescribing.

**4b) Poor infrastructure for families may increase infection risk for patients.**   The team observed few facilities available to families who were supporting their critically ill relative on ICU. The problems included a lack of handwashing and toilet facilities, no dedicated space to wait or sleep, and nowhere to prepare food or wash clothes, either for themselves or the patient. There was often no electrical socket to charge mobile phones either. Instead, the families would congregate on the stairs and hallways outside the ICU in an environment that '*no normal person would ever want to be in*' (Study team observation note). Families even reported having their belongings stolen as they slept.

As well as adding to the hardships that the families had to endure, the lack of facilities was seen to have a knock-on effect for patients. The lack of handwashing facilities in many sites made it hard for the family to comply with infection prevention measures. The lack of somewhere to clean and prepare patient food exposed the patient to an additional infection risk. Being unable to charge mobile phones made it hard to ensure that they were contactable when needed urgently.

**4c) Hostile environment for women.**   The lack of facilities for patients' families posed a particular challenge for women. One young woman, who lived eight hours away from the hospital, spoke of the challenges she faced caring for her husband on the ICU. She described having no access to usable toilet facilities, made worse because she could not '*go outside like the men*'. She described lascivious looks from ICU support staff and from other men both inside ICU and while waiting outside. She described being regularly harassed for money. At night she described seeking out another woman to be next to for safety and even then, she had to stay sitting up all night, getting almost no sleep.

Although some other hospitals in the study had separate toilet facilities for men and women, none of them had dedicated spaces where unaccompanied women could wait in safety.

**4d) Financial hardship impairs patient care.**   All five study sites are government-funded ICUs, where the bulk of care is provided free to patients. The exact details of what patients did and did not have to pay for varied from site to site, but pharmacy expenses were a significant portion of the direct costs reported everywhere. Another source of expenditure was paying the ward support staff who would charge a fee for each service they provided.

*There are many touts here. For example, as the five fingers of the hand are not equal, so the people are here. . . [The support staff] are always looking for money, 'to eat breakfast'. So people change in three shifts from five am, and they look for money, 10 [currency unit], 20 [currency unit]. I say I will not give even one [currency unit]. Why, does the government not give you money? They say, brother, that is not enough. (Interview with a family member, F12)*

Several family members interviewed mentioned distressing interactions with support staff requesting money. One family member suggested that standardising the amounts they charged for each task would improve the situation. Nurses and doctors were aware of the problem, which, in some cases, led to long delays in changing soiled nappies for patients whose families did not pay. In some sites, support staff also positioned themselves as paid intermediaries for procuring investigations.

*When we give a test to get from outside, it is more likely to fall into the hands of the broker. . . [Families] suffer at the hands of brokers. If these can be reduced, it would be better. (Interview with a junior nurse, JN2)*

Patients and families often spoke of the stress and anxiety the hospital costs caused, or, as in this case, the material ramifications:

*We do not have enough money. My mother collects from here and there by borrowing. I and my husband's elder brother are here, we go without eating for two meals, I don't eat. . . . I just eat one meal. I eat one banana at night, bon [a kind of bread], or a cake, just nothing else. I eat whenever and whatever I get. (Interview with a family member, F7)*

Not only did financial hardship impact their ability to fulfil the roles required of families, it also increased the risk that the family would request discharge home against medical advice.

All five sites were linked to a charitable foundation that could provide some form of support for the poorest patients. However, many patients were not aware of its existence. In some of the study sites, staff were much more proactive than others about informing the patients' families and supporting them to access its support. Another difference between the sites was that some ICUs provided a greater range of government-funded drugs and supplies than others. This was seemingly related to how energetic each ICU was in procuring and maintaining a supply of these goods.

## Discussion

In this study we have identified key areas that influence the experiences of critically ill patients and their families in ICUs in Bangladesh and India. We have also identified examples of best practices that can support person-centred care in this setting. Drawing on these findings, our team has compiled a list of recommendations to better support patients and their families on ICU (Table 2). Many of our findings in this study align with evidence from other critical care settings [21]. However, we have also identified additional factors that are important in Bangladesh and India specifically, and may be of relevance in comparable contexts.

Compared to ICUs with a more restrictive visiting policy, a less restrictive ICU visiting policy appeared to improve the experiences of patients and families. Although there is already a strong body of evidence to show the benefits of unrestricted ICU visitation for patients and families [22–24], there is still a wide variation in ICU visiting policies internationally. A survey conducted in 2016 across 47 countries showed that only 35% of respondents said their ICUs were open for family visits twenty-four hours a day [25]. Among respondents in our study, the main opposition to less restricted access was concern regarding infection control. Evidence from a recent meta-analysis shows that this concern may be unfounded [26], although our findings also highlight the need to provide handwashing facilities and education for families to support them to comply with infection prevention and control practices.

Importantly, we found that having a family member present at the bedside enabled the family to contribute to patient care in ways that went beyond providing psychological support. Additional roles of the family included raising the alarm when the patient deteriorated, decreasing the requirement for physical restraint, and supporting mobilization and rehabilitation. The need for the family to fulfil these roles was greatest where ICU staffing levels were lower, suggesting that unrestricted family access to ICU could provide most added value in resource-constrained settings.

To ensure that family members are not expected to perform tasks that are better suited to trained healthcare staff, it would be helpful for ICUs to establish a scope of practice for family members, outlining what should and should not be expected of them. Furthermore, family involvement in care should be entirely voluntary; and where the family are not able not to be involved–or prefer not to be–then the ICU staff should ensure that the patients receive the same quality of service regardless.

Family members should also receive an induction to ICU, plus support and supervision for the tasks they carry out. A successful precedent where families were supported to play a very active role in the care of their critically ill relative was reported during the peak of the COVID pandemic in India [27]. A recent systematic review describes other contexts in which family participation in the delivery of hospital care has proved safe and effective [28].

Despite finding that good communication is a major mediator of patient and family satisfaction, effective and consistent communication with patients and families was extremely challenging in the study ICUs. This echoes the findings of Kumar *et al.* who found that the doctors' lack of time for communication and '*conflicting information about the patient's*

*progress in the ICU being given by different ICU staff* led to confusion and fear on the part of the families [11]. Their study also reported that the doctors were attributed an almost god-like status which inhibited family engagement in decision-making [11].

Based on the example set by two of the study sites where nurses were supported to give routine patient updates to families, we suggest that ICUs explore what information could successfully be relayed by nurses rather than doctors. Some people may prefer to receive all patient updates from doctors [11], but in practice we observed that frequent routine updates from the bedside nurse combined with regular communication from doctors for more complex information was widely accepted by most families.

Across all five study sites, there was a tendency for curative care to be given precedence over symptom management and rehabilitation. A previous study of ICU patients in India noted that they demonstrated 'a high degree of acceptance of the discomfort of injections, bed care and physical restraints in a matter-of-fact way' [29]. This was also evident in our findings and may have influenced pain-relief prescribing practices. Nevertheless, as one study site showcased, good pain management and rehabilitation on ICU is possible in this context. The proven benefits of the ICU liberation bundle, with its emphasis on effective pain management and early mobilisation, has underlined the importance of maintaining an emphasis on these components of care in addition to treating the underlying pathology [7].

Our study linked a failure to meet the physical, environmental and financial needs of families with negative consequences for patients. Specifically, we found that families had to contend with long queues and repeated journeys to-and-fro to procure supplies and investigations, which led to treatment delays for patients. Limited infrastructure for families made it harder for them to safely prepare patient food, wash their hands before entering ICU, or be readily contactable in an emergency. Women faced gender-specific challenges inside and outside the ICU which impaired their ability to support their hospitalised relative. The poorest families struggled to afford essential ICU supplies and could not pay for the services of hospital support staff, with negative sequelae for patient care.

Some of these challenges faced by families have been reported in other studies [8–12, 30, 31]. One study highlighted the inadequate facilities for families, which were described as unclean, overcrowded and lacking in privacy [31]. Another study reports feedback from families who requested a separate ICU pharmacy in the vicinity of ICU [10]. The issue of hospital support staff expecting informal payments for services is explored in more detail in an ethnography by Zaman *et al.* [30]. The authors describe the '*indispensable*' role such staff play in patient care in Bangladesh, while also noting that some individuals treat the patients and families harshly and with disrespect [30]. This mirrors our study finding that the demands of support staff can negatively impact patient and family experience. While difficult to address, this issue evidently warrants further scrutiny.

Although there were no perfect solutions to any of these challenges, we observed that small-scale, locally driven problem-solving can be of benefit. Centring patient and family experiences is a key starting point for exploring solutions. Our findings illustrate how measures to improve family experience could directly lead to improvements in the quality of care the patient receives.

## Strengths and limitations

This is the first study to take a multi-site, comparative approach to examine the experiences of ICU patients and their families–and the factors that influence them–in South Asia. The strengths of the study include the involvement of a multidisciplinary team of researchers with varied professional and sociocultural backgrounds. Data collected through observation

supplemented reports from patients, family members and healthcare staff to generate a rich and nuanced account. Another strength is that the study highlights solutions rather than solely documenting challenges.

Limitations of the study include the sample of interviewees, since families of patients who died or who did not make a good recovery were not invited for interview to avoid distressing them further. Patients who had not recovered sufficiently to be able to consent to participate at the point of discharge from ICU were also not interviewed; nor were hospital support staff. Furthermore, the analysis focused exclusively on events during ICU admission rather than experiences before and after.

## Conclusion

In this multicentre qualitative study, we have identified important mediators of patient and family experience that span many different aspects of care. Factors that promote person-centred care include access to ICU for families, support for family involvement in care delivery, clear communication with patients and families, good symptom management for patients, support for rehabilitation, and measures to address the physical, environmental and financial needs of the family. Our list of practical recommendations based on these findings can support intensive care clinicians and policy makers who wish to implement person-centred care for critically ill patients in similar settings.

## Supporting information

**S1 Checklist. Inclusivity in global health research questionnaire.**
(PDF)

**S1 Text. Sample interviews guides.**
(PDF)

## Author Contributions

**Conceptualization:** Rebecca Inglis, Meghan Leaver, Christopher Pell, Aniruddha Ghose, Mohammad Moinul Islam, Bharath Kumar Tirupakuzhi Vijayaraghavan, Ranjan Kumar Nath, Arjen Dondorp, Swagata Tripathy, Md. Abul Faiz.

**Data curation:** Rebecca Inglis, Christopher Pell.

**Formal analysis:** Rebecca Inglis, Meghan Leaver, Christopher Pell, Suma Ahmad, Mumnoon Ansary, Sidharth B. S., Hasnat Chowdhury, Mohammed Abdur Rahman Chowdhury, Nazmin Akhter Farzana, Sakib Hosain, Subhasish Nayak, Swagata Tripathy, Md. Abul Faiz.

**Funding acquisition:** Rebecca Inglis, Meghan Leaver, Christopher Pell, Arjen Dondorp.

**Investigation:** Rebecca Inglis, Meghan Leaver, Christopher Pell, Suma Ahmad, Shamima Akter, Fakrul Ibne Amir Bhuia, Mumnoon Ansary, Sidharth B. S., Momtaz Begum, Shishir Ranjan Chakraborty, Hasnat Chowdhury, Mohammed Abdur Rahman Chowdhury, Putul Deb, Nazmin Akhter Farzana, Aniruddha Ghose, Mohammad Harun Or Roshid, Md. Rezaul Hoque Tipu, Sakib Hosain, Md. Mozaffer Hossain, Mohammad Moinul Islam, Bharath Kumar Tirupakuzhi Vijayaraghavan, Mohammad Mohsin, Manisha Mund, Shamema Nasrin, Ranjan Kumar Nath, Subhasish Nayak, Nibedita Pani, Shohel Ahmmad Sarker, Swagata Tripathy, Md. Abul Faiz.

**Methodology:** Rebecca Inglis, Meghan Leaver, Christopher Pell, Fakrul Ibne Amir Bhuia, Mumnoon Ansary, Sidharth B. S., Hasnat Chowdhury, Nazmin Akhter Farzana, Sakib

Hosain, Mohammad Moinul Islam, Bharath Kumar Tirupakuzhi Vijayaraghavan, Subhasish Nayak, Swagata Tripathy, Md. Abul Faiz.

**Project administration:** Rebecca Inglis, Meghan Leaver, Christopher Pell, Arjen Dondorp, Swagata Tripathy, Md. Abul Faiz.

**Software:** Rebecca Inglis.

**Supervision:** Meghan Leaver, Christopher Pell, Suma Ahmad, Shamima Akter, Mohammed Abdur Rahman Chowdhury, Mohammad Harun Or Roshid, Md. Rezaul Hoque Tipu, Md. Mozaffer Hossain, Mohammad Moinul Islam, Mohammad Mohsin, Manisha Mund, Ranjan Kumar Nath, Nibedita Pani, Shohel Ahmmad Sarker, Arjen Dondorp, Swagata Tripathy, Md. Abul Faiz.

**Validation:** Christopher Pell, Shamima Akter, Momtaz Begum, Shishir Ranjan Chakraborty, Putul Deb, Aniruddha Ghose, Mohammad Moinul Islam, Bharath Kumar Tirupakuzhi Vijayaraghavan, Mohammad Mohsin, Manisha Mund, Shamema Nasrin, Ranjan Kumar Nath, Nibedita Pani, Shohel Ahmmad Sarker, Swagata Tripathy, Md. Abul Faiz.

**Writing – original draft:** Rebecca Inglis.

**Writing – review & editing:** Rebecca Inglis, Meghan Leaver, Christopher Pell, Suma Ahmad, Shamima Akter, Fakrul Ibne Amir Bhuia, Mumnoon Ansary, Sidharth B. S., Momtaz Begum, Shishir Ranjan Chakraborty, Hasnat Chowdhury, Mohammed Abdur Rahman Chowdhury, Putul Deb, Nazmin Akhter Farzana, Aniruddha Ghose, Mohammad Harun Or Roshid, Md. Rezaul Hoque Tipu, Sakib Hosain, Md. Mozaffer Hossain, Mohammad Moinul Islam, Bharath Kumar Tirupakuzhi Vijayaraghavan, Mohammad Mohsin, Manisha Mund, Shamema Nasrin, Ranjan Kumar Nath, Subhasish Nayak, Nibedita Pani, Shohel Ahmmad Sarker, Arjen Dondorp, Swagata Tripathy, Md. Abul Faiz.

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
