## [Decision Letter · Decision Letter 0]

6 Mar 2024

PGPH-D-23-02475

Understanding patient and family experiences of critical care in Bangladesh and India: what are the priority actions to promote person-centred care?

Dear Dr. Inglis,

Thank you for submitting your manuscript to PLOS Global Public Health. After careful consideration, we feel that it has merit but does not fully meet PLOS Global Public Health’s publication criteria as it currently stands. Therefore, we invite you to submit a revised version of the manuscript that addresses the points raised during the review process.

We look forward to receiving your revised manuscript.

Kind regards,

Manish Barman, MD., MSc., FRCP

Academic Editor

Journal Requirements:

Additional Editor Comments (if provided):

Dear Authors

Kindly address the concerns raised by the reviewers and resubmit a revised version.

Thanks

Manish Barman

Reviewers' comments:

Reviewer's Responses to Questions

**Comments to the Author**

1. Does this manuscript meet PLOS Global Public Health’s publication criteria? Is the manuscript technically sound, and do the data support the conclusions? The manuscript must describe methodologically and ethically rigorous research with conclusions that are appropriately drawn based on the data presented.

Reviewer #1: Yes

Reviewer #2: Partly

Reviewer #3: Yes

2. Has the statistical analysis been performed appropriately and rigorously?

Reviewer #1: Yes

Reviewer #2: N/A

Reviewer #3: N/A

3. Have the authors made all data underlying the findings in their manuscript fully available (please refer to the Data Availability Statement at the start of the manuscript PDF file)?

Reviewer #1: Yes

Reviewer #2: No

Reviewer #3: Yes

4. Is the manuscript presented in an intelligible fashion and written in standard English?

Reviewer #1: Yes

Reviewer #2: No

Reviewer #3: Yes

5. Review Comments to the Author

Reviewer #1: The findings of the study are insightful, however, the study aimed to address three research questions, the presentation of findings for the same can be in line with these questions. Currently the broad themes identified do not clearly address any of these questions, rather a mixture of Q2 and Q3. Also, a diagrammatic representation of the themes identified pertaining to each of these questions can bring out better clarity for the reader.

Reviewer #2: Reviewer Report

Understanding patient and family experiences of critical care in Bangladesh and India: what are the priority actions to promote person-centred care?

PGPH-D-23-02475

General comment

This paper highlights interesting issues in patient-centred care. There is a need for some elaboration in the methods and findings to improve the clarity.

Specific comments

1. Abstract

- Sampling methods should be stated in the methods section

- A summary of recommendations should be stated in the conclusion section

2. Introduction

- Justification on why the study should be conducted in public facilities should be added.

3. Methods

- Please explain the sampling methods of the five hospitals

- Please explain the sampling methods of the patients and care families

- Please elaborate on the questions in the interview guideline

- Please explain the observation checklist, and add it in the supplementary file.

- Please describe the results of coding using thematic analysis

- Did the author use data saturation to define the final number of interviews and observations? Please explain.

4. Results

- Please put the initials of the informant in the quotation, for instance, a senior nurse 1, a senior nurse 2, doctor 1, etc. So that we can have insight that the quotations are coming from various informants.

5. Discussion

- Please discuss whether the findings can be influenced by cultural aspect of people in India and Bangladesh

- Please discuss about the generalisability of the findings in other context in low and middle income countries

6. The article needs English editing

Reviewer #3: Summary:

Overall, the study addressed the lack of evidence on improving patient and family experiences in the intensive care units (ICUs) within the South Asian critical care context, specifically in Bangladesh and India. This research contributed to the understanding of ICU experiences in the South Asian context, offering practical recommendations to enhance person-centered care.

Strengths:

The study demonstrated commendable strength in its comprehensive exploration of critically ill patients', families' and staffs’ experiences in ICUs in Bangladesh and India. The incorporation of international evidence, practical recommendations, and understanding of resource constraints added value to the understanding of person-centered care in diverse healthcare settings. The study's robust methodology, including a reflexive thematic analysis and cross-team collaboration, contributed to its credibility and potential for real-world impact.

Key issues:

a. Study design section:

1. It would be helpful to have a clear justification for why these specific hospitals in Bangladesh and India were chosen. Did they represent a diverse range of patient populations, geographical locations, or other relevant factors?

2. Rationale for Confidentiality: While confidentiality is mentioned, providing a brief rationale for why it's crucial in this study can add depth. For example, explain whether revealing hospital names could lead to potential biases or compromises in the research.

b. Observations:

Overall, the observational methodology appears robust, but addressing the mentioned points could enhance the transparency, credibility of the study.

1. While it is mentioned that oral consent was obtained for observations, more details on the nature of the consent process, including how it was explained to participants and any potential challenges faced, would strengthen the transparency of the study.

2. The clinician researcher who continued clinical work during observations may face challenges in maintaining objectivity. Additional information on how potential biases associated with multitasking were addressed would enhance the credibility of the data.

3. While patients were made aware of the research team's presence when their condition allowed, it would be beneficial to elaborate on how the team managed where patient conditions might have limited their understanding or ability to provide consent?

4. The varying degrees of participation among researchers during observations could introduce potential bias. Further clarification on how this variability was managed, ensuring consistency in data collection?

c. Interviews:

Overall, the interview methodology is well-structured, but addressing the mentioned points could enhance transparency, rigor, and the overall quality of the study.

1. Limited Details on 'Information Power': While the concept of 'information power' is mentioned in determining the target number of interviews, more details on how this concept was practically applied, especially in terms of data saturation, would enhance the methodological transparency.

2. While it is mentioned that one researcher reviewed all transcripts and translations for quality control, additional details on the specific measures taken for quality control, such as inter-rater reliability checks?

3. For interviews conducted in India, where bilingual interviewers translated directly, more details on the process of crosschecking and resolving translation queries could provide insights into the rigor of the translation process.

d. Data analysis:

1. While the coding process is mentioned briefly, providing more detail on how codes were developed, refined, and the criteria for inclusion/exclusion of codes would provide a clearer understanding of the analysis process.

2. The term "recursive manner" is used in describing the generation of preliminary themes. Further elaboration on how this recursive process was conducted would enhance the clarity.

3. It would be beneficial to clarify the role of the clinician researcher who continued to work clinically during observations in the analysis phase. How was potential bias managed in this context?

4. While the involvement of team members in reviewing preliminary themes is mentioned, providing more detail on specific instances of reflexivity and how it enhanced analytical depth would add depth to the methodology.

Minor issue:

a. Study design section: The study's purpose, research questions, or objectives should be stated before the method section and not within the study design.

b. In addition to this, I recommend considering the inclusion of the semi-structured interview questionnaire as a supplementary material or in the main manuscript if space allowed. This addition would contribute to the transparency of the study and provide understanding of the specific questions posed during the interviews. It can also enhance the reproducibility of the research and allow future studies to build upon or replicate the methodology. Additionally, it enables the researchers to evaluate the alignment between research questions, interview questions, and the study's objectives.

c. The discussion section is commendable for its clear identification of key areas influencing ICU experiences in Bangladesh and India. The practical recommendations derived from the study findings are actionable and demonstrate a thoughtful consideration of resource constraints, contributing to the pragmatic impact of the research. I recommend to consider delving further into doctor-patient-family dynamics to enhance the depth.

6. PLOS authors have the option to publish the peer review history of their article (what does this mean?). If published, this will include your full peer review and any attached files.

**Do you want your identity to be public for this peer review?** For information about this choice, including consent withdrawal, please see our Privacy Policy.

Reviewer #1: No

Reviewer #2: No

Reviewer #3: **Yes: **Dimple Dawar

---

## [Decision Letter · Decision Letter 1]

30 May 2024

Understanding patient and family experiences of critical care in Bangladesh and India: what are the priority actions to promote person-centred care?

PGPH-D-23-02475R1

Dear Dr Inglis,

We are pleased to inform you that your manuscript 'Understanding patient and family experiences of critical care in Bangladesh and India: what are the priority actions to promote person-centred care?' has been provisionally accepted for publication in PLOS Global Public Health.

Best regards,

Manish Barman, MD., MSc., FRCP

Academic Editor

Thank you for addressing the suggestions made by reviewers.

Reviewer Comments (if any, and for reference):

Reviewer's Responses to Questions

**Comments to the Author**

1. If the authors have adequately addressed your comments raised in a previous round of review and you feel that this manuscript is now acceptable for publication, you may indicate that here to bypass the “Comments to the Author” section, enter your conflict of interest statement in the “Confidential to Editor” section, and submit your "Accept" recommendation.

Reviewer #1: (No Response)

Reviewer #2: All comments have been addressed

Reviewer #3: All comments have been addressed

2. Does this manuscript meet PLOS Global Public Health’s publication criteria? Is the manuscript technically sound, and do the data support the conclusions? The manuscript must describe methodologically and ethically rigorous research with conclusions that are appropriately drawn based on the data presented.

Reviewer #1: (No Response)

Reviewer #2: Yes

Reviewer #3: (No Response)

3. Has the statistical analysis been performed appropriately and rigorously?

Reviewer #1: Yes

Reviewer #2: N/A

Reviewer #3: (No Response)

4. Have the authors made all data underlying the findings in their manuscript fully available (please refer to the Data Availability Statement at the start of the manuscript PDF file)?

Reviewer #1: (No Response)

Reviewer #2: No

Reviewer #3: (No Response)

5. Is the manuscript presented in an intelligible fashion and written in standard English?

Reviewer #1: (No Response)

Reviewer #2: Yes

Reviewer #3: (No Response)

6. Review Comments to the Author

Reviewer #1: For better understanding of the reader themes can be divided into two broad headings answering the two research questions - i.e. - Current practices that appear to improve experiences for ICU patients and their families, and recommendations to improve ICU experience (can also have current practices as first research question which would be assessed first conceptually). The themes in figure 1 can be in points rather than sentences. Grammar/sentence formation needs to be checked for the themes in fig 1 eg - 2c - afforded or offered? Fig 2 can be described along with current practices.

Reviewer #2: The authors have addressed the reviewer comments adequately.

Reviewer #3: (No Response)

7. PLOS authors have the option to publish the peer review history of their article (what does this mean?). If published, this will include your full peer review and any attached files.

**Do you want your identity to be public for this peer review?** For information about this choice, including consent withdrawal, please see our Privacy Policy.

Reviewer #1: No

Reviewer #2: No

Reviewer #3: No
